# Outcomes in Patients with Pulmonary Arterial Hypertension Underwent Transcatheter Closure of an Atrial Septal Defect

**DOI:** 10.3390/jcm12072540

**Published:** 2023-03-28

**Authors:** Jae-Hee Seol, Se-Yong Jung, Han-Byul Lee, Ah-Young Kim, Eun-Hwa Kim, In-Kyung Min, Nam-Kyun Kim, Jae-Young Choi

**Affiliations:** 1Division of Pediatric Cardiology, Department of Pediatrics, Congenital Heart Disease Center, Severance Cardiovascular Hospital, Yonsei University College of Medicine, Seoul 03722, Republic of Korea; 2Department of Pediatrics, Yonsei University Wonju College of Medicine, Wonju 26426, Republic of Korea; 3Division of Pediatrics, Soonchunhyang University College of Medicine, Seoul 04401, Republic of Korea; 4Biostatistics Collaboration Unit, Department of Biomedical Systems Informatics, Yonsei University College of Medicine, Seoul 03722, Republic of Korea; 5Department of Thoracic and Cardiovascular Surgery, Severance Cardiovascular Hospital, Yonsei University College of Medicine, Seoul 03722, Republic of Korea

**Keywords:** atrial septal defect, pulmonary arterial hypertension, pulmonary artery systolic pressure

## Abstract

Pulmonary arterial hypertension (PAH) related to an atrial septal defect (ASD) poses a challenge to transcatheter closure of an ASD (tcASD). We aimed to determine the predictors for remaining PAH (rPAH) post-tcASD. This retrospective study was conducted at a single tertiary university hospital. Adult patients with an ASD and PAH were divided into three groups according to pulmonary vascular resistance (PVR). Normalization of pulmonary atrial systolic pressure (PASP) was defined as an estimated right ventricular systolic pressure < 40 mmHg and was determined using transthoracic echocardiography. Among 119 patients, 80% showed PAH normalization post-tcASD. Normalization of PAH post-tcASD was observed in 100%, 56.2%, and 28.6% of patients in mild, moderate, and severe PVR groups, respectively. The patients’ New York Heart Association functional class improved. Multivariate logistic regression analysis showed that age and high PVR were significant risk factors for rPAH. A receiving operator curve analysis showed a PASP cutoff value > 67.5 mmHg to be predictive of rPAH post-tcASD, with an area under the curve value of 0.944 (sensitivity, 0.922; specificity 0.933). Most patients, including moderate-to-severe PAH patients, improved hemodynamically and clinically with tcASD. Since patients with severe PAH are at a risk of rPAH, tcASD should be performed by selecting the patient carefully based on pre-procedure medication, a vasoreactivity test, and a balloon occlusion test.

## 1. Introduction

Atrial septal defect (ASD), a commonly diagnosed structural heart lesion occurring in approximately 2 per 1000 live births [1], accounts for 25–30% of all adult congenital heart diseases [2]. In patients with untreated ASDs, chronic pulmonary over-circulation due to shunt flow can cause pulmonary vascular remodeling and increased pulmonary vascular resistance. These changes may result in atrial arrhythmias, right ventricular dilation, right heart failure, and pulmonary hypertension [3]. Zwijnenburg et al. reported that, in 30 studies published before March 2017, the prevalence of PAH ranged from 29% to 73% in adult patients with ASDs and from 5% to 50% post-ASD closure [4].

Transcatheter closure of ASD (tcASD) in patients with pulmonary arterial hypertension (PAH) improves PAH severity and cardiac functional capacity, and reduces atrial arrhythmias [5]. However, some patients show remaining PAH (rPAH) or aggravation of PAH post-ASD closure. PAH is a strong predictor of mortality in older patients who undergo ASD closure [6,7]. Hence, the decision to opt for ASD closure should be carefully considered in high-risk patients with PAH.

According to the European Society of Cardiology and the European Respiratory Society 2020 guidelines for adult congenital heart disease, ASD closure is indicated for adults with PAH who exhibit a pulmonary vascular resistance index (PVR) value < 3 WU/m^2^ but is contraindicated at a PVR > 5 WU/m^2^ [8]. Additionally, as per the American Heart Association/American College of Cardiology 2018 guidelines, ASD with elevated pulmonary artery pressure (PAP) and pulmonary vascular resistance (PVR) more than two-thirds systemic is considered to be a contraindication for closure [9]. However, it is difficult to determine the outcomes for ASD closure in patients with moderately-to-severely elevated PVR, as cut-off PAP or PVRi levels have not been established for these patients; several studies have reported that for the efficacy of PAH medication and subsequent transcatheter closure (treat and repair strategy) in patients with severe PAH, a cut-off PAP for tcASD after PAH medication remains unclear [10,11,12].

Several studies have reported a decrease in the prevalence of PAH post-tcASD; however, they have not identified risk factors for rPAH [13,14,15,16,17,18]. This study aimed to identify predictors of tcASD outcomes in patients with PAH.

## 2. Methods

### 2.1. Patient Selection

We retrospectively surveyed 1593 patients with ASD who had undergone ASD closure from May 2005 to December 2017 at the Division of Pediatric Cardiology, Severance Cardiovascular Hospital, Yonsei University Health System. Among them, 119 (74%) adult patients (>18 years old) had PAH (PAP, ≥25 mmHg) but no evidence of Eisenmenger syndrome. The patients were divided into three groups based on pulmonary vascular resistance (PVR), using cardiac catheterization, as follows: mild (PVR, ≥3 and <5 WU), moderate (PVR, ≥5 and <7 WU), and severe (PVR, ≥7 WU) [19,20].

### 2.2. Personalized Treatment Strategies

The indication for closure of ASD included right-side heart enlargement and Qp:Qs > 1.5:1. In moderate-to-severe PAH patients, transient balloon occlusion test was performed. In the case of an increase in left atrial or pulmonary capillary wedge pressure to >10 mmHg, fenestrated device was used. Decline in cardiac output and/or desaturation by balloon occlusion test were considered contraindications for ASD closure [21,22]. Additionally, in some patients, a vasoreactivity test with high O_2_ (10 L/min through a mask) and/or aerosolized iloprost for 15 min was also performed [23]. For patients with persistent AFib post-closure, antiarrhythmic drugs were used according to our hospital strategy. Radiofrequency catheter ablation (RFCA) was performed for rapid ventricle rhythm refractory to medical therapy. If there was hemodynamic instability, direct-current (D/C) cardioversion was performed [24].

### 2.3. Follow-Up

Physical examination, 12-lead ECG, chest radiography, and echocardiography were performed at 1 week, at 1, 3, 6, and 12 months, and annually thereafter. Patients with moderate-to-severe PAH were regularly followed up every 3 to 6 months. Normalization of PAH was defined as an estimated PASP < 40 mmHg on echocardiography. Mild, moderate, and severe PAH were defined as an estimated PAH of 40~49 mmHg, 50~59 mmHg, and ≥60 mmHg at last follow-up, respectively. The occurrence of major cardiac events (MACEs) during the follow-up period was also investigated as an outcome of post tcASD. Death, stroke or hospital admission due to cardiac-associated disease were defined as MACEs.

### 2.4. Statistical Analysis

We used the ANOVA test and independent sample t-test for patient comparisons. We performed Kaplan–Meier survival analysis with a log-rank test to compare the time to normalization between the groups. Univariate and multivariate logistic regression analyses were performed to identify risk factors for rPAH. Finally, we calculated a receiver operating characteristic (ROC) curve to determine a cut-off PAP value for PAH normalization. All statistical analyses were performed using the statistics program SPSS ^®^ (Version 25, IBM, Chicago, IL, USA), and results with *p*-values < 0.05 were considered statistically significant.

### 2.5. Ethics Statement

The Institutional Review Board of Severance Hospital approved this study (study approval number: 4-2021-0925). The requirement for informed consent was waived due to the study’s retrospective nature.

## 3. Results

### 3.1. Ethics Statement

The study population included 119 consecutive adult patients with ASD and PAH who underwent successful tcASD, and for whom no major complications occurred during the follow-up period. Baseline characteristics are summarized in Table 1. Of 119 patients, the prevalence rates of PAH in patients with mild, moderate, and severe PAH were 61.3% (*n* = 73), 26.9% (*n* = 32), and 11.8% (*n* = 14), respectively.

The average age was relatively low in the severe group (*p* = 0.0049). Female patients outnumbered male patients, and the association between female sex and PAH was stronger than that between male sex and PAH (*p* = 0.0479). As the severity of PAH increased, the functional capacity (New York Heart Association Class (NYHA)) was limited (*p* < 0.0001). No significant differences were observed in ASD size and tricuspid regurgitation (TR) according to PAH severity. The Qp/Qs was lower in the severe PAH group.

### 3.2. Treatment Outcomes Post-tcASD

Treatment outcomes post-tcASD are shown in Figure 1. In patients with moderate-to-severe PAH, tcASD was performed if there was a response to the balloon occlusion test and/or vasoreactivity test. In all mild PAH patients, pulmonary pressure was normalized after tcASD. In the patients with moderate PAH, 18 (56.3%) patients were normalized. Among 14 (43.8%) patients with rPAH, 12 patients improved to mild PAH, and the other 2 had worse or similar PASP. In patients with severe PAH, only 4 (28.6%) patients were normalized, but 5 of 10 patients with rPAH showed a decrease in pulmonary pressure.

Hemodynamic and clinical changes post-tcASD are shown in Table 2. The median follow-up period was 53 months (interquartile range (IQR) 23.8–82.7), 12.1 months (IQR 29.8–66.1), and 51.4 months (IQR 40.7–76.5) for the mild, moderate, and severe groups, respectively. Based on the high degree of agreement between PASP and pre-procedural RVSP in our study (ICC, 0.8426), post-closure PASP was estimated by measuring RVSP using echocardiography. Normalization of PASP was seen in 100%, 56.2%, and 28.6% of the patients in the mild, moderate, and severe groups, respectively (*p* < 0.0001). The median time to normalization was 1.68 months (IQR 0.43–8.21), 11.04 months (IQR 5.29–26.51), and 39.79 months (IQR 25.49–76.70) in the mild, moderate, and severe groups, respectively, and significantly longer in the severe group (*p* < 0.001). Patients with severe PAH were associated with late normalization. In the severe PAH group, 28.5% (*n* = 4) of the patients remained in severe PAH (*p* = 0.0015), with two having worsening PAH. The severity of PAH was associated with the development of MACEs during the follow-up period. (*p* < 0.0001) There was no significant difference in the incidence of post-procedural AFib according to the severity of PAH. However, 14 of 28 patients with AFib converted to normal sinus rhythm post-tcASD. Among them, two patients immediately post-tcASD, three patients after D/C cardioversion during the procedure, three patients after RFCA, and one patient were converted to normal sinus rhythm while using antiarrhythmic drugs (Appendix A). Most patients showed improvement in NYHA class. Overall, there was a decrease in the proportion of patients with NYHA class III–IV and an increase in NYHA class I, which was particularly significant in the severe group (Appendix A).

A treatment flow chart for patients with severe PAH is shown in Figure 2. Among patients in the severe group, five (35.7%) patients received selective pulmonary vasodilation pre-tcASD. Of these, three (21.4%) patients required a fenestrated device. After tcASD, severe PAH remained in two patients, and the other patient improved to moderate PAH. Two patients who did not require a fenestration device remained as severe PAH. Of the nine patients who started medication after direct device closure, four patients were normalized. The other five patients, including two patients who used the fenestrated device, improved to mild-to-moderate PAH. The pulmonary vasodilator was maintained except for those who were normalized. Information on post-tcASD medication and treatment outcomes in patients with severe PAH are summarized in Appendix A. Except for 1 patient who died of gastro-intestinal bleeding, 4 of 13 patients were able to stop taking the pulmonary vasodilator after normalization. Patients who improved with mild PAH were maintained on only a single medication. Dual medication was maintained for moderate PAH, and dual or triple medication was maintained for patients with severe PAH. Two of five patients with AFib returned to normal sinus rhythm after RFCA, and the other three patients were maintained on antiarrhythmic drugs to manage AFib. PASP was reduced after tcASD in most patients, except for one patient who experienced increased PASP (Appendix A). Most patients had transient NT-proBNP elevations post-procedure, but the latest follow-up results showed a decreasing trend (Appendix A). When clinical outcomes were compared, all 13 patients showed improvement in the 6 min walking distance test. However, in terms of NYHA class, while six patients improved, the rest worsened or remained similar to pre-procedure measurements (Appendix A).

Figure 3 shows the Kaplan–Meier curves of the proportion of patients with normalization of PASP over time post-tcASD according to the severity of PAH. During the median follow-up period post-tcASD, patients with mild-to-moderate PAH showed normalization of PAH relatively earlier post-tcASD compared to patients with severe PAH (*p* < 0.0001 using log-rank analysis).

### 3.3. Factors Associated with PAP Normalization

A comparison of the baseline characteristics and hemodynamic data between the groups with normalized PASP and rPAH is shown in Table 3. Of the 119 patients, 95 (80%) patients were normalized, and the other 24 (20%) patients had rPAH. Among the hemodynamic variables, the independent factors associated with normalization were baseline PASP (*p* < 0.0001), mPAP (*p* < 0.0001), Qp/Qs (*p* = 0.0272), PVR (*p* < 0.0001), and NYHA class (*p* < 0.0001). Compared to the rPAH group, the normalized patients had higher Qp/Qs (2.19 vs. 2.46, *p* = 0.0005), lower PVR (6.45 vs. 3.4, *p* < 0.0001), and relatively low median PASP (45, (IQR 40–60 mmHg) vs. 70 (IQR 70–100) mmHg) at baseline. Another factor associated with rPAH was the NYHA class (*p* < 0.0001). Patients with NYHA class III or higher were more likely to remain as PAH. Patients with rPAH were more symptomatic pre- and post-tcASD (*p* < 0.0001). Baseline characteristics such as age, weight, body surface area (m^2^), and sex, were not significantly associated with rPAH. Moreover, TR, ASD size, and pre-and post-procedure Afib, which indicate the degree of heart disease, were not related to postoperative normalization. MACEs occurred more in rPAH patients.

Univariate analysis showed that higher PASP (*p* < 0.0001), lower Qp/QS (*p* = 0.0306), higher PVR (*p* < 0.0001), and a more severe grade of TR (*p* = 0.040) were correlated more closely with rPAH (Table 4). Further analysis was performed with stepwise selection. Multivariate analysis showed that for a 1-year increase in age, there was a 1.1-fold increase in the risk of rPAH (OR 1.10, 95% CI 1.03–1.14, *p* < 0.0024). In addition, as the PVR increased by 1, rPAH risk increased 4.6-fold (OR 4.60, 95% CI 2.53–8.36, *p* < 0.0001) (Table 5).

ROC curve analysis showed that the best cut-off PASP value with which to predict rPAH post-tcASD was 67.5 mmHg, with an area under the curve (AUC) of 0.944 (sensitivity 0.922, specificity 0.933; Appendix A).

## 4. Discussion

Pulmonary hypertension can occur in the presence of ASD and can lead to increased mortality in cases of unrepaired ASD [25]. In ASD patients, PAH initially develops dependent on the left to right shunt, and as pulmonary vascular remodeling progresses gradually, after a certain point, the PAH is no longer dependent on only the shunt. Some patients may develop PAH post-tcASD with similar pathophysiology to idiopathic pulmonary hypertension. Although there are guidelines for ASD closure indication, in patients with ASD and moderate-to-severe PAH, it is difficult to predict the risk for rPAH before ASD closure as the guidelines for such patients are ambiguous. In a recent Nyboe et al. [26] cohort study and a Selai et al. [18] meta-analysis, ASD closure was found to improve clinical outcomes, including survival rate, pulmonary pressure, and right ventricular function in patients with PAH. In addition, there was a study that reported successful results of ASD closure through strategies using “treat and repair” [10] and “fenestrated device” [27] in patients with severe PAH. Appropriate intervention can reduce pulmonary pressure and improve quality of life for patients with PAH, even severe PAH, when it is possible to select patients who could benefit from ASD closure. While several studies have established the criteria for ASD closure in these patients, a consensus has not yet been reached.

This study aimed to present our experience of tcASD in patients with PAH and clarify the risk of rPAH post-tcASD by comparing outcomes following tcASD according to PAH severity. Most patients with mild-to-moderate PAH achieved normalization of PASP post-tcASD. In patients with severe PAH, although PASP decreased significantly post-tcASD, 28.6% of them remained as severe PAH, and 64% of them required a continuous selective pulmonary vasodilator. The strong factor predicting rPAH was high PVR before the procedure. In addition, our study indicated that patients with a PASP > 67.5 mmHg were likely to have rPAH post-tcASD based on the ROC curve. These findings were consistent with previous studies that analyzed PAH outcomes post-tcASD [13,20]. Our results also support the clinical effect of ASD closure in patients with moderate-to-severe PAH. As estimated using echocardiography, Doppler examination findings indicated a significant reduction in PASP at the time of the last patient follow-up post-tcASD. Moreover, we observed an improvement in NYHA class post-tcASD, even in patients with severe PAH. This improvement was due to a volume reduction in the right side of the heart and increasing the systolic volume of the left ventricle [28]. 

In the analysis of baseline characteristics, younger age and female sex were associated with PAH severity. Younger and/or female patients had higher PASP at baseline. This finding indicated that the duration of PAH may not affect PAH severity, as other studies have reported [6]. However, rPAH was associated with age in our results. As age increased by 1 year, the OR of rPAH was 1.1. There were two relatively young (in their 20 s and 30 s) patients without any comorbidities who experienced worsened PAH post-tcASD. Furthermore, our study showed a trend towards higher prevalence and severity of PAH in women. These findings suggest that other additional factors such as an underlying genetic predisposition or hormonal differences may be associated with PAH development. Further research involving more patients with severe PAH is needed.

Contrary to expectations, AFib and severe TR did not show a significant association with PAH severity and rPAH. Elevated PAP increases pressure in the right atrium and causes right atrial enlargement, which is a risk factor for atrial arrhythmias, such as AFib [26]. Our findings showed that pre and postprocedural AFib were not significantly associated with PAH severity and rPAH. TR, which often occurs secondary to pulmonary hypertension, can cause right ventricular remodeling and tricuspid annulus dilatation. Thus, TR progression is associated with severity of PAH and poor prognosis for PAH. However, in our study, TR severity was not related to PAH severity and rPAH.

This study shows the possibility of ASD closure in patients who were previously reluctant to undergo ASD closure due to severe PAH. Patients with severe PAH required a variety of treatment strategies. A “treat-and-repair” strategy could be an option. In addition, by observing changes in pulmonary pressure through the vasoreactivity test or balloon occlusion test during diagnostic catheterization, patients can be carefully selected for tcASD. Therefore, while device closure can be actively considered in patients with mild-to-moderate PAH, tcASD should be determined carefully for patients of old age and with severe PAH. In our study, balloon occlusion and pulmonary vasoreactivity tests were used to predict pulmonary pressure reversibility in patients with moderate-to-severe PAH; however, further studies are needed to develop a unified standard for patients with PAH. 

In our study, eight patients underwent tcASD with a fenestrated device. All fenestration in device was spontaneously closed for about a month, indicating that the effect of fenestration was ambiguous. Further study should be performed to evaluate the specified effect of the fenestrated device.

## 5. Study Limitations

Our study has limitations that warrant consideration. First, our study was a retrospective study. We could not include all patients with ASD with PAH. The results may reflect a selection bias in that only patients who underwent tcASD were observed, excluding patients who underwent surgical closure and failed ASD closure. Second, the number of patients included in the study was small. Since all results of patients with ASD with PAH could not be represented, there were limitations in deriving results for predicting rPAH after tcASD. Third, we could not perform follow-up catheterization post-tcASD. In most cases, clinical symptoms improved after closure, so invasive evaluation such as catheterization was not performed in consideration of the risks and benefits. Since catheterization and follow-up PVR evaluation were performed in only a small number of patients, statistical comparison analysis could not be conducted. However, there is a limitation in evaluating rPAH only by echocardiogram after tcASD in that echocardiographic parameters are affected by TR and RV function.

## 6. Conclusions

In our study, risk factors for rPAH after procedure were age and high PVR. For ASD closure in patients with severe PAH, tcASD should be determined carefully and an approach through various treatment strategies is required. A “treat and repair strategy” might be an option. In addition, the patient should be carefully selected by the observation of PVR change through vasoreactivity and balloon occlusion tests, and then closure should be considered. For patients with a predictable poor prognosis, research on the risk assessment of ASD closure in patients with PAH will be needed for a more individualized treatment plan.

Our study showed that percutaneous suture ASD is feasible and may have substantial clinical benefits, even for patients with moderate-to-severe PAH. A prospective and randomized study is needed to clarify the indicators for ASD device closure indication in patients with increased risk.

## 7. Patients

Patients and/or the public were not involved in the design or conduct or reporting of this study.

## Figures and Tables

**Figure 1 jcm-12-02540-f001:**
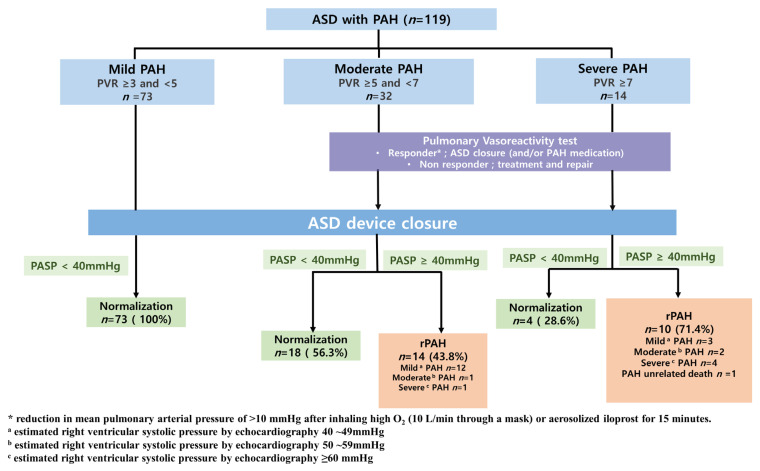
Flow chart of treatment and outcomes according to PAH severity in ASD patients. ASD, atrial septal defect; PAH, pulmonary arterial hypertension; PVR, pulmonary vascular resistance; PASP pulmonary atrial systolic pressure.

**Figure 2 jcm-12-02540-f002:**
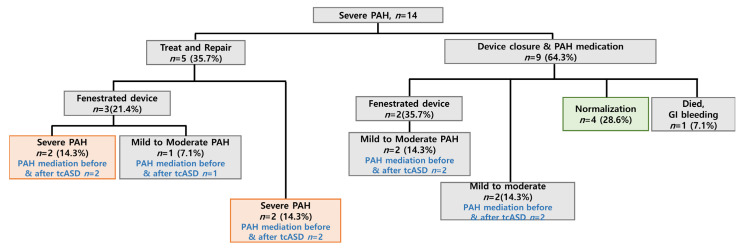
Treatment flow chart for patients with severe PAH. GI, gastrointestinal; PAH, pulmonary arterial hypertension.

**Figure 3 jcm-12-02540-f003:**
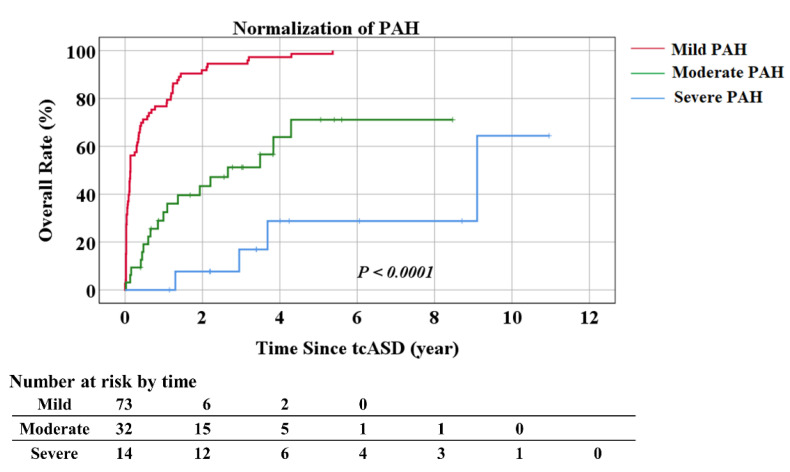
Kaplan–Meier curves showing the cumulative rate of normalization of PAH according to pulmonary arterial systolic pressure. PAH, pulmonary arterial hypertension; tcASD, transcatheter closure of atrial septal defect.

**Table 1 jcm-12-02540-t001:** Baseline characteristics of patients with ASD according to the degree of PAH.

Variables	Group			*p*-Value
	Mild (*N* = 73)	Moderate (*N* = 32)	Severe (*N* = 14)	Overall
Age (years) at cath	48.85 ± 13.03	50.48 ± 17.78	35.74 ± 13.47	0.0049 *
Wt (kg)	61 (53, 70)	54.5 (49.6, 63.0)	51.5 (48.0, 57.0)	0.0139 *
Ht (cm)	161.68 ± 9.98	159.14 ± 7.74	160.50 ± 6.26	0.4159
BSA (m^2^)	1.68 ± 0.20	1.59 ± 0.17	1.58 ± 0.14	0.0369 *
Sex				0.0479 *
Male	23 (31.5)	8 (25.0)	0 (0.0)	
Female	50 (68.5)	24 (75.0)	14 (100.0)	
Pre-procedural AFib	13 (17.8)	10 (31.3)	5 (35.7)	0.2360
ASD_size (mm)	26.34 ± 6.39	26.16 ± 6.77	24.29 ± 7.03	0.5604
TR > GII	10 (13.7)	6 (19.4)	2 (14.3)	0.7596
MR > GI	13 (17.8)	7 (22.6)	1 (7.1)	0.4559
Pre-procedural echo RVSP	43 (40, 52)	59.5 (55.0, 67.0)	89.5 (69.0, 110.0)	<0.0001 *
PASP (mmHg)	45 (40, 50)	60.0 (51.0, 64.5)	85.5 (74.0, 100.0)	<0.0001 *
mPAP (mmHg)	25 (25, 30)	34 (30, 40)	50 (42, 60)	<0.0001 *
Qp/Qs	2.54 (2.13, 3.29)	2.40 (1.93, 2.83)	1.67 (1.40, 2.28)	0.0009 *
PVR (WU)	3.32 (3.15, 3.48)	5.56 (5.30, 6.21)	8.45 (8.17, 9.12)	<0.0001 *
NYHA				<0.0001 *
1	43 (58.9)	6 (19.4)	1 (7.1)	
2	25 (34.2)	21 (67.7)	7 (50.0)	
3	5 (6.8)	4 (12.9)	5 (35.7)	
4	0 (0.0)	0 (0.0)	1 (7.1)	

AFib, atrial fibrillation; ASD, atrial septal defect; BSA; body surface area; GI, grade I; GII, grade II; mPAP, mean pulmonary arterial pressure; MR, mitral regurgitation; NYHA, New York Heart Association functional class; PASP, pulmonary arterial systolic pressure; PVR, pulmonary vascular resistance; Qp, pulmonary flow; Qs, systemic flow; RVSP, right ventricular systolic pressure; TR, tricuspid regurgitation. * *p*-value < 0.05 was considered to be statistically significant.

**Table 2 jcm-12-02540-t002:** Hemodynamic and clinical outcomes post-tcASD.

		Group		*p*-Value
Variables	Mild (*N* = 73)	Moderate (*N* = 32)	Severe (*N* = 14)	Overall
Last follow-up RVSP	32 (30, 33)	36.0 (31.0, 40.5)	50.5 (38.0, 69.0)	<0.0001 *
Normalization	73 (100)	18 (56.2)	4 (28.6)	<0.0001 *
Time to normalization (months)	1.68 (0.43, 8.21)	11.04 (5.29, 26.51)	39.79 (25.49, 76.70)	<0.0001 *
Persistent severe PAH	2 (2.7)	1 (3.2)	3 (21.4)	0.0274 *
Follow up months	53.62 (23.77, 82.68)	42.05 (29.84, 66.05)	51.44 (40.70, 76.47)	0.5870
MACEs †	6 (8.2)	12 (37.5)	8 (57.1)	<0.0001 *
Post-procedural AFib	6 (8.2)	5 (15.6)	3 (21.4)	0.2720
Last NYHA				<0.0001 *
1	57 (78.1)	17 (54.8)	6 (42.9)	
2	15 (20.5)	13 (41.9)	4 (28.6)	
3	1 (1.4)	1 (3.2)	4 (28.6)	
Pul.vasodilator at last follow-up	0 (0.0)	17 (53.1)	10 (71.4)	<0.0001 *

PAH, pulmonary arterial hypertension; RVSP, right ventricular systolic pressure; tcASD, transcatheter closure of atrial septal defect. † Total MACEs 26 cases: 1 death, 1 stroke, 24 admissions (11 admissions for heart failure, 7 pulmonary. HTN aggravation, 6 arrhythmia). * *p*-value < 0.05 was considered to be statistically significant.

**Table 3 jcm-12-02540-t003:** Comparison of baseline variables between groups with normalization and rPAH (normalization of PAH is defined as an estimated right ventricular systolic pressure < 40 mmHg).

Variables	Normalization		*p*-Value
	No (*n* = 24)	Yes (*n* = 95)	
Age (years)	58.66 (33.49, 69.41)	47.64 (35.93, 57.82)	0.1285
Weight (kg)	52.05 (48.85, 61.70)	59.00 (52.00, 69.00)	0.0484 *
Height (cm)	159.33 ± 8.06	161.24 ± 9.29	0.3566
BSA (m^2^)	1.55 (1.46, 1.64)	1.62 (1.53, 1.80)	0.0471 *
ASD-size (mm)	24.95 ± 7.70	26.33 ± 6.22	0.3623
Sex			0.5146
Male	5 (20.8)	26 (27.4)	
Female	19 (79.2)	69 (72.6)	
TR > GII	7 (29.2)	11 (11.7)	0.0525
MR > G1	4 (16.7)	17 (18.1)	>0.999
PASP (mmHg)	64.0 (54.5, 77.5)	45.0 (40.0, 59.0)	<0.0001 *
mPAP (mmHg)	39 (29, 50)	27 (25, 30)	<0.0001 *
Qp/Qs	2.19 (1.58, 2.74)	2.46 (2.07, 3.13)	0.0272 *
PVR (WU)	6.45 (5.40, 8.45)	3.40 (3.20, 4.85)	<0.0001 *
Device_size	28.0 (22.0, 32.5)	30.0 (26.0, 34.0)	0.2433
Pre-procedural AFib	9 (37.5)	19 (20.2)	0.0980
AFib at last follow-up	5 (20.0)	9 (9.6)	0.1500
MACEs	10 (40.0)	16 (17.0)	0.0130 *
NYHA			<0.0001 *
1	2 (8.3)	48 (51.1)	
2	13 (54.2)	40 (42.6)	
3	8 (33.3)	6 (6.4)	
4	1 (4.2)	0 (0.0)	
Last NYHA			<0.0001 *
1	7 (29.2)	73 (77.7)	
2	12 (50.0)	20 (21.3)	
3	5 (20.8)	1 (1.1)	

AFib, atrial fibrillation; ASD, atrial septal defect; BSA, body surface area; MR, mitral regurgitation; NYHA, New York Heart Association functional class; MACEs, major adverse cardiovascular events; mPAP, mean pulmonary arterial systolic pressure; PASP, pulmonary arterial systolic pressure; PVR, pulmonary vascular resistance; Qp, pulmonary flow; Qs, systemic flow; TR, tricuspid regurgitation; RVSP, right ventricular systolic pressure; tcASD, transcatheter closure of atrial septal defect. * *p*-value < 0.05 was considered to be statistically significant.

**Table 4 jcm-12-02540-t004:** Univariate analysis for rPAH post-tcASD.

Variables	OR	95% CI Lower	95% CI Upper	*p*-Value
Age (years)	1.025	0.994	1.058	0.1149
Weight (kg)	0.984	0.955	1.013	0.2778
Height (cm)	0.976	0.928	1.027	0.3542
BSA (m^2^)	0.080	0.006	1.117	0.0604
Sex				
1: Male	1 (ref)			
2: Female	1.432	0.485	4.230	0.5162
ASD_size	0.968	0.904	1.037	0.3599
Device_size	0.952	0.890	1.018	0.1485
PASP	1.074	1.039	1.110	<0.0001 *
Qp/Qs	0.475	0.242	0.933	0.0306 *
PVR (WU)	2.610	1.811	3.760	<0.0001 *
TR > GII	3.107	1.053	9.165	0.0400 *
MR > G1	0.906	0.274	2.993	0.8712
pre-procedural AFib	1.768	0.820	6.233	0.1790
Post-procedural AFib	1.365	0.750	4.27	0.2130

AFib, atrial fibrillation; ASD, atrial septal defect; BSA, body surface area; CI, confidence interval; MR, mitral regurgitation; NYHA, New York Heart Association functional class; OR, odds ratio; PAH, pulmonary arterial hypertension; PASP, pulmonary arterial systolic pressure; PVR, pulmonary vascular resistance; Qp, pulmonary flow; Qs, systemic flow; rPAH, remaining PAH; RVSP, right ventricular systolic pressure; TR, tricuspid regurgitation; tcASD, transcatheter closure of atrial septal defect. * *p*-value < 0.05 was considered to be statistically significant.

**Table 5 jcm-12-02540-t005:** Multivariate logistic regression analysis of rPAH post-tcASD.

Variables	OR	95% CI Lower	95% CI Upper	*p*-Value
Age	1.114	1.051	1.180	0.0003 *
PVR	4.605	2.548	8.323	<0.0001 *

PVR, pulmonary vascular resistance. * *p*-value < 0.05 was considered to be statistically significant.

## Data Availability

The data used to support the findings of this study are available from the corresponding author upon request.

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
