# Peer review of "Outcomes in Patients with Pulmonary Arterial Hypertension Underwent Transcatheter Closure of an Atrial Septal Defect"

_jcm, 2023, doi:10.3390/jcm12072540_

Round 1
Reviewer 1 Report
Dear Authers
1- The introduction provides sufficient context. It outlines what Atrial Septal Defect (ASD) is, why it is important to consider when diagnosing and treating it, and what the current guidelines are for treating it. It also outlines the purpose of the study, which is to identify the predictors for the outcome of transcatheter closure of ASD in patients with pulmonary arterial hypertension.
2- The method provides sufficient detail.
3- This article presents a study into the effects of transcatheter atrial septal defect (tcASD) closure in patients with pulmonary hypertension (PAH). The authors found that most patients with mild-to-moderate PAH achieved normalization of pulmonary artery systolic pressure (PASP) post-tcASD. In patients with severe PAH, the PASP decreased significantly post-tcASD, but 28.6% of them remained as severe PAH, and 64% of them required a continuous selective pulmonary vasodilator. The authors also found that younger age and female sex were associated with PA
4- Moderate English changes required
5- Could you please provide an update on more recent research in this area?
Regards
Author Response
We appreciate your systematic review of our study. Your comments have encouraged us to improve this study and add more valuable information. Thank you very much.

Reviewer 2 Report
Hee Seol, J. and colleges performed a retrospective analysis of the predictors for remaining PAH (rPAH) after transcatheter atrial defect closure (ASD) using a cohort of 119 patients who underwent ASD percutaneous closure regardless of the values of pulmonary vascular resistance.
The manuscript is well written, the concepts are clearly exposed and the figures are very clear. However, I have some suggestions to include in the manuscript as well as some limitations and conclusions which I think should be emphasized.
This is a small cohort of patients since the 60% of the patients did not have increased risk of persistent pulmonary hypertension, reducing the sample at risk to less than 50 patients. So the detection capacity of predictors of pulmonary hypertension is limited and this should be addressed in the limitations section. Moreover when there is not hemodynamic data after the procedure and systolic pulmonary pressure is dependent of RV systolic function and its echo quantification is limited in presence of severe TR (more common in late stages of this pathology).
Although it is a small sample, these patients were out of the guidelines and even though, the authors found that percutaneous closure ASD is feasible and may have a real clinical benefit for the patient regardless of pulmonary vascular resistance. I would have emphasized this aspect in the conclusions as well as the real need for randomized studies.
In this sense, I miss the number of patients with failed occlusion test who were excluded for the percutaneous treatment and the surgical cohort.
Minor observations
Line 54 “Additionally, as per the American 54 Heart Association/American College of Cardiology 2018 guidelines, ASD with elevated pulmonary artery pressure (PAP) and pulmonary vascular resistance (PVR) less than two-thirds systemic is considered to be a contraindication for closure”. It should be stated “more than two-thirds”.
Line 58 “theoutcomes” space missing.
Author Response

(The authors gave the same response as above.)
